# Psychosocial Functioning and the Educational Experiences of Students with ASD during the COVID-19 Pandemic in Poland

**DOI:** 10.3390/ijerph19159468

**Published:** 2022-08-02

**Authors:** Aneta Lew-Koralewicz

**Affiliations:** Institute of Pedagogy, University of Rzeszów, 35-010 Rzeszów, Poland; alew@ur.edu.pl

**Keywords:** pandemic COVID-19, autism spectrum disorder, emotional well-being, social functioning, online education

## Abstract

Due to their individual developmental and learning needs, adolescents with autism spectrum disorder (ASD) benefit from a variety of educational, medical, and therapeutic services. During the COVID-19 pandemic, these services were discontinued or significantly reduced, which may have resulted in increased difficulties in coping with various areas of life. The purpose of this study was to explore how the pandemic affected the psychosocial and educational functioning of students with ASD. A qualitative, problem-focused interview method was used. The obtained material was subjected to interpretive phenomenological analysis. The study involved 10 secondary school students diagnosed with ASD. The assessment of the effects of the pandemic on the functioning of people with ASD is inconclusive. The respondents noted both negative and positive effects of lockdown. On the positive side, they were able to spend time with their family, isolate themselves from difficult social relationships and feel better. Among the negative effects, adolescents point to difficulties in emotional functioning—increased levels of stress and anxiety, as well as increased feelings of loneliness and difficulties with online education. The study showed the varied experiences of young people with autism during the pandemic, highlighting the significant need to support some of them in terms of their emotional, social and educational functioning.

## 1. Introduction

The global spread of the SARS-CoV-2 virus and the high mortality rate from coronavirus (COVID-19) prompted the World Health Organization to declare a pandemic in March 2020. Its impact was felt around the world, significantly affecting different areas of life [1]. The coronavirus (COVID-19) pandemic was a major threat to the human life and health. One significant impact of the pandemic is its effect on the mental health of the population resulting in stress, isolation, reduced social contact and access to the support system [2,3,4]. In addition to significant economic problems [5,6,7], the pandemic caused threats to public health, safety, and well-being of individuals and communities. Most societies experienced insecurity, confusion and emotional isolation resulting from the closure of schools, jobs and public institutions. The difficult situation during the pandemic and limited access to medical care can result in a range of emotional responses such as anxiety and mental problems, as well as problematic behavior [8,9]. Online medical services often did not meet the needs of patients and made it difficult to provide professional support [10]. A Polish study confirmed decrease in the level of happiness and satisfaction with life during the COVID-19 [11], pandemic resulting in a negative impact on mental health [11,12,13].

This situation has had a significant impact on the well-being and mental health of young people [14,15,16,17]. Some adolescents, particularly girls, experienced a significant increase in symptoms of depression and anxiety, as well as a marked decrease in satisfaction with life [18,19]. Concerns related to COVID-19, difficulties with online learning, and increased conflict with parents, as well as limitations to social interactions due to isolation, were all predictors of mental health problems of adolescents [18,20]. The pandemic had an impact on their educational situation, including a deterioration in the educational performance of pupils in several countries [21]. Changing the form of education for those with an increased need for support could be a challenging. The assessment of this form of education is ambiguous. Young people in Poland stressed both the advantages and disadvantages of distance learning [22]. The advantages included the ease of using digital tools and support from parents and teachers [23]. Among the disadvantages of online learning were difficulties in focusing, organizing self-study and self-discipline, loss of engagement, negative emotions (stress, fatigue, fear of passing exams), and lack of direct contact with peers [23,24,25,26]. In general, students rated the quality of remote education worse than traditional education [23].

Considering that people with ASD are a notable group in society and that the prevalence of autism spectrum disorder has increased significantly [27,28], it is reasonable to assume that a large group of individuals with ASD will need support and assistance, especially in the difficult situation during the pandemic. Clinical features characteristic of autism spectrum disorders such as restricted, repetitive patterns of behavior which often manifest themselves as insistence on sameness, inflexible adherence to routines, or ritualized patterns of behavior [29], are symptoms that can cause various difficulties in coping with the changes brought about by the pandemic [30]. The pandemic had a significant impact on the quality of life of individuals with autism and their families [30,31,32,33]. It is important to note that individuals with ASD are at elevated risk for psychiatric problems, particularly due to their high pre-pandemic rates of psychiatric comorbidities and the disruption during lockdown to daily routines and access to support [34,35]. Children with pre-existing behavioral problems, such as autism and attention deficit hyperactivity disorder, are highly likely to have clinical deterioration. Anxiety, depression, irritability, boredom, inattention, and fear of the virus are prevalent psychological problems in individuals during the COVID-19 pandemic [36].

The significant problem experienced by individuals with ASD was isolation resulting from the closure of schools and treatment facilities [37,38,39,40,41].

Previous studies of the potential effects of pandemic on the psychological well-being of adolescents with ASD are contradictory, as it shows both negative and positive impacts of the pandemic on the psychological well-being of people with ASD [33,42,43,44,45,46,47]. Studies conducted in many countries have indicated the significant changes that occurred in the functioning of many people with ASD during the pandemic. The COVID-19 pandemic had a wide-ranging impact on the behaviors of children with ASD, and created challenges for their caregivers in the USA. Researchers identified several significant factors that predict greater difficulties for the child, including pre-existing behavioral challenges, disrupted sleep and a diagnosis of depression [16]. Families with children on the spectrum reported greater behavioral problems during the lockdown, as well as more parental distress [33]. The findings of Charalampopoulou et al. show that more than half of autistic children experienced a deterioration in their mental health [37]. The psychological well-being of adolescents and their parents declined overall under the 2020 partial lockdown in Switzerland. The results are contradictory depending on variables such as age, ASD severity, or features of the family [17]. The results of the Nisticò et al. study conducted during the first and second lockdowns in Italy, indicate that individuals with HF-ASD experienced higher levels of stress, anxiety, depression, PTSD-related symptoms, tiredness, and perceived wellbeing during the second lockdown in comparison to the first one. However, they also reported to feel subjectively more comfortable and less tired during the lockdown than before, in relation to the social distancing measures [48]. A similar finding was also reported in Saliverou et al. study [44]. In Poland, there has been little research on the situation of people with ASD during the pandemic. A large-scale study was conducted by Pisula et al., who found that in the group of those with ASD, 29% of anxiety symptoms and just over 46% of depression symptoms reached the clinical levels. In addition, levels of anxiety and depression were higher in those with higher levels of autistic traits, and as well as in those who felt lonely during the pandemic [49]. People with ASD in Poland also experience a sense of loss due to the lack of direct contact with others. On the other hand, people with ASD expressed their satisfaction with mandatory social distancing [43]. Other studies have focused on the educational situation of students with ASD. The biggest problems students with ASD experienced were comprehension and memorizing of new material, as well as communication. During online education, interaction with their peers with ASD worsened, in addition to difficulties in communicating with their teachers and peers [38].

Todate, there has been little systematic research exploring the psychological impact of the COVID-19 pandemic on adolescents in Poland. Therefore, the current qualitative study considered how restriction affected the psychosocial functioning of adolescents with ASD in Poland. The aim of this study is to explore how adolescents with autism make sense of their personal experiences of COVID-19, especially their individual perception of social and educational functioning as well as their psychological well-being during the pandemic. Therefore, the research question of this study is: How has the COVID-19 pandemic impacted on overall functioning of adolescents with ASD in Poland? It is expected that the results of this study will help to identify the difficulties faced by adolescents with ASD in order to promote effective support in difficult situations such as COVID-19.

## 2. Methods

### 2.1. Design

The study was based on Interpretative Phenomenological Analysis (IPA), a qualitative research method. This approach aims to explore how people make sense of their experiences. An IPA study involves intensive and detailed analysis of the accounts produced by a comparatively small number of participants, who are purposively selected [50]. IPA is recognized as an effective qualitative approach in autism research. This is because the purpose of IPA research is to provide in-depth analyses of the experiences of a specific group, and the research is conducted with homogeneous, small samples. IPA also assumes that research participants are treated as experts and attempts to find a balance between autistic informants and non-autistic researchers in the analysis of participants’ individual experiences and feelings [50]. A characteristic of IPA is reflexivity, which demands from researchers to consider the impact of their own experiences and preconceptions on research design and procedures [51].

### 2.2. Participants

The study included 10 high school students diagnosed with autism spectrum disorder. The majority of respondents (*n* = 9) were male [M1–M9] and one female [F1]. The age of the participants ranged from 16 to 18 years (M 16.7, SD 0.78). The cognitive development of seven participants was normative, while three of them had mild intellectual disabilities [M2, M3, M9]. Some of the participants had other comorbid mental health conditions: obsessive-compulsive behaviors [M7] or anxiety disorders [F1], with one of the boys [M8] having tic disorders. Three of them [M2, M7, M8] received pharmacological treatment before the pandemic. All of them were raised in intactfamilies. Two of the participants were an only child [M8, M9], while the other participants had siblings with typical development. The participants were students from different state schools that were closed during the pandemic. All respondents studied remotely during the lockdown and had online lessons delivered live. During the pandemic, (between March 2020 and March 2022) students were educated remotely for 12 months, while for 8 months they were educated in schools. However, individual classes were subject to quarantine, which shortened the period of face-to-face education. The Polish education system offers psychological and pedagogical support for students with ASD, but the pandemic has made it difficult for them to benefit from this form of intervention. During the pandemic some of them received psychological and pedagogical support remotely. Two participants received this support at school [F1, M8]—consisting of one class per week. Two adolescents declared that they did not receive this kind of intervention during the pandemic [M4, M6]. Participants were deprived of access to support in therapeutic centers for the first four months of the pandemic, and thereafter received therapy and support with varying frequency, from once a month to once a week. In addition, access to psychiatric care was significantly reduced during the pandemic. Selection for the research group was via nonprobability sampling, and the criteria for selection were diagnosis of autism spectrum disorder, developed communication competence, and attendance at a state high school.

The participants’ level of functioning corresponds to level 1 according to the DSM-5. Study participants were recruited by their therapist, who determined whether they met the study’s inclusion criteria. Of the 15 families selected by the therapists, 11 agreed to participate in the study, and one participant did not meet the inclusion criteria, so 10 participants ultimately took part in the study.

### 2.3. Procedure

Ethical approval was granted by the Bioethics Committee at the University of Rzeszów. Participants were recruited from an association dedicated to helping people with developmental disabilities. The study was conducted at the therapy center the participants attended so as to ensure a sense of security and to avoid making changes that might increase feelings of anxiety. The study was conducted at institutions cooperating with the university, and the study participants had prior contact with the researcher, which made it easier to establish closer contact and create an atmosphere of openness during the interviews. Semi-structured interviews were conducted by the author so as to allow the researcher to explore in depth the experiences of the study participants [51]. The researcher has clinical experience in the diagnosis and therapy of people with ASD. Meetings were conducted using direct face to face contact, with adherence to the necessary safety procedures in force during the pandemic. Demographic data was collected at the beginning of the meeting. Then the researcher asked open-ended questions through a semi-structured interview format [52]. The schedule included four open-ended questions to encourage participants to talk about their experiences during the pandemic (What does pandemic mean to you? How the COVID-19 pandemic has affected your school life? Could you describe your relationship with peers during the pandemic? Could you tell me about your feelings, emotions during the pandemic?). The interviews were conducted at the therapeutic facility attended by the respondents. After receiving all the information about the study, parents and adolescents consented to participate in the study and to be recorded. The meetings lasted between 25 and 40 min, depending on the participant’s involvement. No incentive was offered, but participants received a university goody bag after the interview.

### 2.4. Data Analysis

The collected research material was transcribed, followed by interpretive phenomenological analysis, using MAXQDA software (version 22.0.1, VERBI GmbH, Berlin, Germany). The analysis was conducted using the principles presented by Smith et al. and included reading and re-reading, coding, clustering, iteration, narration and contextualization [53]. Participants’ names and any identifying information were coded to preserve anonymity and confidentiality. In analyzing the text, the author focused on the subjective perspectives of the participants. In-depth inductive qualitative analysis was conducted to reveal the adolescent’ unique perspectives on handling the pandemic, through a detailed examination of their individual perceptions and life experiences. Based on the data analysis, three superordinate themes were identified, each comprising several subthemes, which are described in the following section.

## 3. Results

The study participants experienced the pandemic in various ways. Due to the fact that the pandemic had a significant impact on different spheres of life, these ranged from emotions experienced, through social relationships to educational aspects. The experience of the pandemic had both positive and negative dimensions and varied greatly between individuals. Data analysis gave rise to three superordinate categories, that is: emotional functioning, social functioning, and functioning as a learner. For each of the categories, a set of subcategories also emerged, relating to positive and negative experiences during the pandemic.

### 3.1. Emotional Functioning

The first superordinate theme concerns the emotions and feelings that all participants experienced, and included such subthemes as stress, anxiety, confusion, sadness, loneliness, and loss of motivation and psychological well-being. The pandemic and its accompanying restrictions evoked different emotions among people around the world. For adolescents with ASD, who are a particularly vulnerable group, the difficult period of the pandemic may have led to significant emotional changes. For some respondents, the pandemic was a stressful time in their lives. This is confirmed by the descriptions of their well-being during the lockdown:

“I don’t feel safe and I’m nervous more often.”[M8]

“I’m more nervous and angrier, sometimes it’s hard for me to control my emotions.”[M5]

“I’m very nervous. I can’t sleep because I keep thinking about it.”[F1]

One of the most frequently experienced feelings was anxiety, which was mentioned by the majority of the respondents. This anxiety was associated with the virus itself, as well as with the possibility of infection and disease. Sometimes it was also the fear of one’s own death or the death of close family members. Frequent changes in pandemic restrictions were also a cause of anxiety. The study participants described anxiety in the following ways:

“I feel anxiety again, very strong anxiety. I’m tired of this pandemic. I’m afraid of the virus, and I’m afraid of death. (…) I’m afraid I’ll die or someone in my family will die. And I don’t want to get sick because I’m afraid of it.”[F1]

“I was afraid of this virus because you don’t know if you’re going to die or not, I couldn’t handle it because I’d never experienced it, before in my life.”[M6]

“I’m definitely scared more often than I was before the pandemic. I haven’t had time to get used to it yet, and there are already new changes.”[M7]

“I am afraid of this virus and I want it to stop, I don’t want to be infected because many people have died.”[M9]

Some of the respondents had disturbed thought patterns due to the pandemic, with one of them experiencing compulsive behaviors in the form of constantly washing and sanitizing his hands, and checking the number of infections and deaths several times a day.

“I look on the internet for news about COVID to see how many cases and how many deaths there are in each province. I check it several times every day to see how many there are in our area.”[M7]

The study participants also pointed to a sense of confusion due to the difficulty in dealing with information chaos, constant change and the lack of logic to some restrictions. The lack of transparency, clear rules and guidelines, caused the anxiety and confusion that some respondents experience. Constancy and predictability are very important to individuals on the autism spectrum. If these needs cannot be met, they can have a destructive effect on their feeling of security and emotional well-being. One participant said:

“My parents try to explain to me what’s going on, I talk to my mom, or read the news on the Internet, but reading doesn’t help, on the contrary, it makes my thinking even more chaotic. It’s hard to put it all in order. There are a lot of contradictions and unknowns. The information is inconsistent and you can get lost in it all and there are no clear guidelines because it all changes and you have to adapt all the time.”[M8]

Some participants reported that a feeling of sadness was an accompanying emotion. Sadness sometimes resulted from reduced contact with others such as family, peers, friends, and supportive therapists:

“It makes me sad because I don’t go to classes and I don’t meet my therapists and I don’t have anything to do.”[M2]

Students with ASD also experienced feelings of sadness related to having to cancel holiday plans or trips. Another adolescent reported that:

“Sometimes I am sad because I would like to go somewhere and I don’t know if we will go on vacation this year, because if they introduce some restrictions again we might not go anywhere.”[M7]

Occasionally, respondents were unable to identify the reasons for their mood, but indicated that it worsened significantly during the pandemic period, as demonstrated by the following statement:

“I am also probably sadder and more depressed, my mood is definitely worse.”[M5]

Many of the respondents were unable to contact their extended family during the pandemic, especially their grandparents. Those separated from parts of their family were accompanied by feelings of longing.

“I miss my grandma and grandpa because I can’t go to them as often. I used to visit them all the time, but now everyone is scared of the virus and it’s better not to infect them, I think.”[M4]

“We were afraid of infecting grandma and I missed her a lot.”[M5]

Some respondents with ASD experienced an increased sense of loneliness during the pandemic. Feelings of loneliness were particularly noticeable in the statements of indyviduals with high-functioning autism who have a significant need for social relationships. School attendance normally allows them to maintain contact with their peers but the lack of such opportunities exacerbated the difficulties they faced in establishing relationships, as illustrated by the respondents’ statements bellow:

“But the worst part is that I don’t have much contact with my friends.”[M4]

“All in all I have no ideas what to do to make it better, I feel a bit lonely.”[M5]

“It’s hard to be alone, maybe if it wasn’t for the pandemic it would be easier to live and I wouldn’t be alone.”[M6]

It is also important to note that for some, feelings of loneliness are not associated with lockdown, but are a constant emotion they experience, which accompanied them both before and during the pandemic. One participant described his situation in this way:

“Nothing has changed, before the pandemic I had no friends either and I was alone.”[M7]

Some participants reported that their motivation to be active had decreased and they felt a general reluctance to engage in activity. This decrease in activity relates to both daily responsibilities and educational aspects. Respondents indicate a sense of meaninglessness, as well as worrying about the current situation. These symptoms in some adolescents may indicate a risk of depressive disorder.

“I don’t want to get up and I have a constant mess in my room and missed homework.”[M4]

“Actually I stopped going out of the house, at the beginning I was afraid, but now I’ve got used to it and I don’t go out alone, I’ve limited these outings to a minimum. When I’ve done everything, I don’t have anything to do, I have such a sense of pointlessness.”[M8]

“I get a little bored. Sometimes I lie down and don’t want to do anything, I just worry about what will happen next. I can’t stop thinking about it.”[F1]

An extreme case was a man who was hospitalized as a result of emotional problems caused by the pandemic:

“Well, it’s much better now, but you know I was in a psychiatric clinic. You know, I couldn’t handle it, but anyone could have failed due to this pandemic, anyone, right? Anyway I, for example, couldn’t cope, because, you know, it hit me like that, I couldn’t stand it.”[M6]

In addition to unpleasant emotional experiences, many people notice improvements in their mental health during the pandemic. They were calm, content and generally felt better. Students who experienced positive effects of the pandemic, particularly lockdown, are those who have significant difficulties with social relationships and are reluctant to interact with others. Going to school is a stressor for them, and during the pandemic they had the opportunity to reduce the negative, unpleasant stimuli they face in their regular school day. Staying in the home environment allowed them to calm down and stay relaxed, increasing their sense of security, which helped to improve their emotional functioning.

“In fact, it’s nice if you don’t have to go to school, I was rather glad to stay home. Mom says I seem to be calmer. Well I get less nervous, sometimes only during lessons. But I think I prefer staying at home.”[M2]

“But the coolest thing was that I could relax at home and it was great when everybody was there, well when mom didn’t go to work. Sometimes dad stayed too, and that was good, because you can’t always have that kind of time, I guess that’s the good thing about this whole pandemic.”[M1]

“I’ m calmer and I don’ t get angry anymore because I’ m less tired and there’ s not so much noise and nobody touches me or says things I don’ t want to hear. I like it a lot, I sit at home and I don’ t have to go to school, I just have my lessons on the computer and classes too, remedial classes and normal classes.”[M9]

“I think I am more peaceful and feel better, because I like the house and quietness.”[M3]

According to the study analysis, the pandemic was a stressful experience for a significant number of respondents. Common feelings accompanying people with ASD during the lockdown were anxiety, confusion, sadness, longing and loneliness. Decreased motivation to be active, worrying, and a sense of meaninglessness in life were disturbing to them. A worse mood and decreased motivation to be active may indicate an increased risk of depression. However, for some people, lockdown provided a respite from the daily demands and difficulties of social relationships, which contributed to improved emotional well-being.

### 3.2. Social Functioning

Deficits in social development are a clinical symptom of autism spectrum disorders, and the pandemic significantly altered the form of interpersonal interactions. One category that emerged from the study was family relationships. In their statements, respondents noted the importance of their immediate social group—the family environment—in coping during lockdown. Adolescents who felt comfortable in their families were more likely to point out positive aspects of the pandemic. In well-functioning families, adolescents with ASD had an increased feeling of security and were able to strengthen their relationships with relatives, which was definitely a positive effect of the pandemic. Some participants described their positive experience:

“My mom works more at home and it’s cool because she’s with me more often and we can do something fun together. We spend more time together.”[M3]

“It’s good to have my parents at home because they help me out and I like it when they are there.”[M1]

“I like being at home with my mom. Mom used to work from home and we could do different things together, we used to play and it was great. Mom used to help me with my lessons online.”[M2]

However, those in whose families relationships were sometimes problematic noted that some of the difficulties of having to stay together all the time were exacerbated. The pandemic affected all members of the family system, which in some cases intensified intrafamily conflicts.

“And that’s the problem. I have no peace in my free time because my younger sister is always bothering me. She gets bored too, and she comes and is malicious, and we often quarrel, and I need peace and quiet.”[M5]

“The worst part was staying at home. You know, I don’t get on well with my mom and sister, and I used to be able to go out, but when the whole lockdown started I thought I couldn’t stand it anymore. My sister argues and picks fights about everything, my mom supports her and is also always saying something to me all the time, but I can’t listen to it anymore and it’s getting worse.”[M6]

For teenagers in adolescence, peers are an important reference group. Peer relationships are a deficient, problematic area in individuals with ASD. During isolation, adolescents with ASD were significantly restricted in their ability to relate to their peers. Their feelings about this area are illustrated by the following statements:

“I miss going out and interacting with my friends. At school there were more of them.”[M5]

“Well, I don’t really have anyone to hang out with, because everyone has their own thing going on, you know, studying or something.”[M6]

Students indicated that they missed peer contact, even if it had not been very close before the pandemic. Neurotypical peers compensated for the lack of direct contact through remote forms, social networking, or other forms of online contact. Students with ASD used these modalities to a lesser extent:

“Well now I don’t really interact with my peers, just sometimes on Teams, but I don’t interact much.”[M7]

“Online contact is rather rare, and when it does occur, it is of a formal rather than a social nature. It usually relates to homework topics, or the implementation of joint projects. I rather write very little. If I need to do a project together then yes, but socially I rather do not write.”[M5]

“I don’t write to anyone, only to teachers and the teacher for remedial education, but at school they rarely talk to me anyway.”[M9]

Students point out that they do not have enough competence to build satisfactory, remote contact with their peers, although they feel the need such contact. Their lack of skills makes it difficult for them to engage in a relationship, and the pandemic has exacerbated these difficulties. One respondent describes their unsuccessful attempts to contact people:

“And I don’t have a chance to meet my friends, and I can’t even go to the neighbors. I wrote to them, but their mother wouldn’t let me write anymore because I sent them some bad content.”[M4]

One important category that is an indicator of social development is leisure activities. Analysis of respondents’ statements indicates that respondents with ASD tend to spend time on solitary activities, sometimes together with their parents. The most common leisure time activities include playing computer games, using the Internet, watching TV, going for a walk, and less often, sports or reading books. For the vast majority of people, these are solitary activities. Lack of contact with others hinders stimulation of social development.

“I was looking for activities to give me some exercise and keep me going. I would watch movies and TV shows, build a little bit with Lego—projects like that. Sometimes mom would come up with some cool activities when she was home.”[M3]

“I just lie down or sleep. I like to sleep a lot. Sometimes I watch something on Youtube, or play some games on the computer. Sometimes I watch TV series. Sometimes I go into the field and play with my dog.”[M2]

Some people felt bored, which resulted from an inability to organize their time with regard to their previous habits.

“I was terribly bored because you couldn’t go anywhere.”[M1]

“Well, it was kind of boring when you couldn’t go out anywhere.”[M3]

The pandemic contributed to a reduction in social interactions among the students interviewed. It is important to note that some individuals did not experience significant changes in social functioning due to low levels of participation in social interactions prior to the pandemic.

### 3.3. Functioning as a Student

The pandemic significantly changed the conditions of education in Polish schools. The respondents studied in state high schools as part of inclusive education, which meant that most of the time they participated in online education. They were asked to express their views on this form of education and assess both the advantages and disadvantages of online education. Pupils who have significant difficulties in functioning in a peer group were satisfied with this form of classes.

“I have better grades and teachers have more time for me. I like to study on Teams. Well, the lady sends tasks and the lessons are a little shorter. I prefer to study on the computer, it’s cool, nobody annoys me and the teacher talks to us a lot about different subjects.”[M7]

“I like to study on the computer, not always, but it’s better than going to school. And it’s more fun to do homework.”[M2]

Part of the respondents found both positives and negatives sides of online learning:

“I’m used to it by now. Sometimes it’s even more fun, but sometimes Teams mess up a little bit and I get terribly nervous when we have a test whether I can get it done on time and send it to the teacher. But sometimes I can’t concentrate as others are messing around. I have to submit a lot of written work on time.”[F1]

Some pupils expressed dissatisfaction with remote education, pointing out the difficulties they experience in the process. These relate to the lack of support during classes, problems with concentration or lack of motivation to participate in lessons.

“I prefer going to school, because online learning was boring and I had to do everything with my mother, because it was hard for me to study by computer. At school the teachers explain things better and the classes are more interesting.”[M3]

“I don’t like online learning very much. I’ve already got a lot of F’s. Well, when my sister or my dad helps me it goes better, but when I’m alone it’s hard for me to keep up with the classes and I don’t always understand what the teachers say. My grades are bad now and I have to improve them, I have to. But I don’t know how, because I keep getting distracted and I don’t know how to do it. At school someone watches over me, but here I have to watch myself, I have to!”[M4]

Online learning has given students more opportunity to avoid participating in classes. Students with low internal motivation admitted to not participating in parts of the classes that did not interest them or which caused them difficulties.

“I’ll tell you a secret, sometimes I don’t log on to the classroom, I just play a game or watch a TV series. And when nobody is watching me, I sort of avoid these on-line lessons a bit. But you can also log on and do something else, but I don’t tell my parents about it.”[M4]

“Sometimes I just log in and pretend as if I don’t have the Internet, but that’s when I don’t want to listen.”[M1]

“If I don’t like something I don’t log on or I don’t say anything and I say that the Internet wasn’t working and sometimes I don’t pay attention, I don’t concentrate.”[M2]

Online learning was perceived by some students as a preferred form of classes; however, these are the opinions of students who were able to handle online classes independently. Those who were less able to cope with their educational responsibilities emphasized a lack of the support that they needed, which caused stress and also a decrease in motivation to participate in on-line lessons.

## 4. Discussion

Results of his study indicate that both positive and negative consequences can be observed in the psychosocial functioning of people with ASD during the pandemic. The ambiguous impact of lockdown has been confirmed by a number of studies in which individuals with ASD as well as their families indicate both positive effects related to the reduction of social demands and normalization of daily life, as well as a negative impact of the pandemic on their mental health [48,54,55,56,57,58].

Analysis of the research results shows that the pandemic was a stressful experience for a significant number of respondents. The stress worsened their well-being and resulted in increased tension and feelings of difficulty in mental functioning. Other studies [47,48,57,58,59] reported similar findings, confirming that increased stress in situations of isolation was experience by many respondents with ASD.

Common feelings experienced by people with ASD during the pandemic were anxiety and at times depressive symptoms. A disturbing phenomenon was the loss of motivation to be active, worrying and losing any meaning of life. According to other studies, nearly one third of people with neurodevelopmental disorders experienced mild or more severe symptoms of anxiety and depression [60]. Increased feelings of anxiety were also found in the general population [61], however, due to the specificity of clinical symptoms, people with ASD are a particularly vulnerable group. Compared to individuals with typical development, adolescents with ASD experienced higher levels of anxiety and symptoms of depression [49]. The impact of the pandemic on some individuals with ASD may be associated with a poorer mood and an increased risk of depression, with higher rates in women with ASD [54]. Obsessive-compulsive behaviors or phobias were a problem during the pandemic, and intensified in some respondents, as evidenced by other studies [44,62].

Loneliness and longing was a common experience of adolescents with ASD during the pandemic, especially during the strict lockdown period. Similar trends were seen in a study by Pellicano and colleagues [63]. Feelings of loneliness for some individuals did not change during the pandemic, as they admitted that for them it was daily experience. However, it is important to note that higher levels of loneliness increase the risk of anxiety and symptoms of depression [49].

Individuals with ASD were at high risk of experiencing problems in functioning during the COVID-19 pandemic. Previous research confirms that adolescents with ASD experience increased psychiatric problems, which is related to children’s understanding of COVID-19, COVID-19 illness in the family, low family income, and elevated parental depression and anxiety symptoms [34]. Individuals with ASD were therefore at high risk of problems in functioning during COVID-19. These issues included aggressive behavior, self-injury, hyperactivity, problems with sleep, eating disorders, pain sensitivity, pain sensitivity, and impaired concentration [64,65,66,67].

The timing of the pandemic significantly changed the social functioning of many individuals with autism, as isolation contributed to reducing contact among the adolescents studied. This is supported by research that has shown significant reductions in support, participation in daily activities, and socialization [36], which intensifies feelings of social isolation [68]. Some of the participants became more closed in on themselves and limited contacts with their environment, for example by staying at home, which they treated as a safe place. Saliverou and team came to similar conclusions in their study, where it was shown that individuals with ASD accepted the safety of the home as an appropriate environment in which they did not experience emotional, psychological, and social difficulties [44].

Family environment determinants also significantly affected the well-being of the respondents. The positive side of this period was the strengthening of family relationships for some individuals. In families where respondents received support, this contributed to their well-being and influenced their positive perceptions of lockdown and the opportunities to spend time with family members. In families where relationships were not supportive, respondents reported increased problems with daily life. Positive emotions were more often expressed in families where the child was having difficulty in school and felt most secure at home. For these individuals, self-isolation and social distancing contributed to their feeling comfort, and were, a time for relaxation and calm, also an escape from the social demands they have to face on a daily basis [69]. These trends are supported by previous studies, which reported that a supportive family is an important factor in protecting children from stress and is a predictor of their well-being [70]. Family characteristics are one of the determinants of the psychosocial effects of social isolation for individuals with ASD [47,71].

In addition to functioning in the family environment, relationships with the peer group are an important indicator of social functioning. Young people emphasize the value they place on direct social contact with close friends [72]. As the results of this study show, adolescents and young adults with ASD perceive the loss of social contact as difficult [73]. Adolescents with ASD also had difficulty compensating for their lack of relationships using forms of remote contact. Most made limited use of online networking opportunities. As other studies have shown, online communication was a solution only when there was prior experience with such communication opportunities. People with ASD most often want to establish friendly interactions, but they have great difficulties in this area [44], which is why they very often choose solitary activities in their free time.

However, participants also indicated positive effects related to a reduction in some social demands and the normalization of daily life [58]. For individuals who experience relationship difficulties, the marked decrease in social interactions increased their sense of well-being, which may be explained by the clinical features of ASD [48].

The change in conditions for studying and the use of distance learning was another challenge for respondents with ASD. Satisfaction in this area among young people in the general population was mixed, and for many, online education was frustrating. The problems faced were lack of consultation, poor organization of learning, the effort needed to learn, technical difficulties and insufficient opportunities to use the software [74]. Individuals with high-functioning autism but with high adaptive skills, performed well during online education and were satisfied with this form of learning [44,75]. Online education was less beneficial for students with ASD with lower levels of cognitive development. They indicated greater difficulties in online education [75] and difficulties in using computers [44].

The study has several practical implications. The findings provide the perspective of adolescents with ASD, which allows researchers, clinicians and teachers to better understand the needs of this group. The research also allowed the support team to gain a deeper insight into what young people with ASD experience. This helped to identify people with aggravated problems in emotional functioning in order to organize support adequate to their needs. Parents, teachers and healthcare staff should strive to work together to compensate for most of the disrupted basic services and support. The results show the necessity of organizing support in order to create social interaction under pandemic conditions. From the perspective of social functioning, efforts are needed to restore social connections with friends, class peers and teachers [68]. In the absence of other options, it is advantageous to use online contact [43]. The results of this investigation may inform interventions in reducing the negative impact of COVID-19 on individuals with ASD.

There are some limitations that should be considered when analyzing the results of this study. The study did not reflect the daily life adolescents with level 2 and 3 autism spectrum disorder (according to the DSM-5). In addition, the study did not cover the situation of adolescents from single-parent families.

## 5. Conclusions

The results of the study show that people with ASD are very different in terms of their experiences during the pandemic. The individual experience of this situation depends on a number of personal and social factors. Individual factors include level of cognitive development, social competence, personality characteristics, and mental health prior to the pandemic, all of which had a significant impact on coping during the lockdown. Individual factors interact with those of a social nature such as family situation, an individual’s position in their group, and opportunities for social support. These aspects determine the respondent’s sense of well-being and coping during the pandemic. The results indicate the need to support individuals with ASD during isolation. Interventions are needed to support the sense of well-being, aimed at preventing the worsening of emotional difficulties and, in some cases, mental disorders.

## Data Availability

The qualitative data that support the findings of this study are available on request from the corresponding author. The data are not publicly available due to privacy and ethical or legal restrictions.

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
