# Peer review of "Psychosocial Functioning and the Educational Experiences of Students with ASD during the COVID-19 Pandemic in Poland"

_ijerph, 2022, doi:10.3390/ijerph19159468_

Round 1

Reviewer 1 Report

Thanks to teh author for this interesting study.

My comments are:

The introduction explains the context and the purpose of the study but needs some extra information on the author background experience and roles.

It will be useful for the author to comment on the level of functioning of participants before the pandemic.

97- More details on participants are needed to help readers judge if they are similar to their patients.

Were participants attending school five days a week, were they attending any after school clubs/activities.

Were participants in receipt of specific educational support at school and/or at home. Do participants have other comorbid mental health conditions (i.e. ADHD, OCD, epilepsy, tics, challenging behaviors or self-harm etc..) or were they in receipt of medication or have any other extra support i.e. local mental health services/therapists or voluntary organizations/charities.

Do the state schools were participants attended have any provisions for ASD. How long were schools of participants closed for during the pandemic.

Did participants receive any educational materials at home, were online lessons or programs delivered live or recorded. Did participants had to do any exams during the pandemic.

103- what is a complete family, more details will help. Also do the family of participants have more than one member with ASD

118- the participants were interviewed in a third facility (not home or school) and the author did not comment on the effects of this on participants as it might have caused them some additional anxiety being a new setting.

119- how many individuals did the author approach before getting the 10 participants. Was there any incentive offered?  Were participants known to the author before the study.

204- saddler?

Discussion; this section could do with a revision. The author mentions some new information in here with references when the focus should be on the author’s interpretations of the study results, limitations and factors that have contributed to the results positively or negatively  

495- this better placed in the discussion section

497- this is a redundant statement

Author Response

Dear Reviewer,

I thank you for a supportive review and comments which may increase the quality of the manuscript.

Reviewer 2 Report

Dear Author,

Congratulations for this hard work.

You can find my recommendations in the attached file.

I wish conveniences.

Author Response

Dear Reviewer,

I would like to thank you for the positive feedback and helpful comments.

Reviewer 3 Report

Thanks for opportunity to review manuscript entitled ‘‘Psychosocial Functioning and the Educational Experiences of Students with ASD during the COVID-19 Pandemic in Poland’’ for International Journal of  Environmental Research and Public Health. Overall, although the article is generally well written and deserves to be published in this journal some necessary and minor revisions must be made before the publication of the article. The strength of the manuscript includes examining variables of interest in a country where such studies are scarce and examining an interesting topic that need more scientific studies. Because my main philosophy of reviewing a manuscript as reviewer and sometimes an editor to improve the manuscript and not punishing the authors, I provided very specific and detailed peer review of the manuscript to increase its quality and citation potential. I hope author of the manuscript may benefit from my review. Necessary revisions reported section by section with the page and line number and when possible with suggestions.

Title

1. No need revision.  

Abstract

2. Page 1, Line  18-19: Conclusion and practical implication sentences are missing in the abstract. Author must add a sentence regarding to practical implications of study findings and a conclusion sentence.

Introduction

3. Page 1, Line 40-41: The citation/citations needed for following sentence ‘ ‘This situation has had a significant impact on the well-being and mental health of young people.’’

4. Page 1-2, Line 41-43: The citation/citations needed for following sentence ‘ ‘Some adolescents, particularly girls, experienced a significant increase in symptoms of depression and anxiety, as well as a marked decrease in satisfaction with life’’.

5. Page 2, Line 43-45: Among who? Subject missing in the following sentence ‘ ‘Concerns related to COVID-19, difficulties with online learning, and increased conflict with parents, as well as limitations to social interactions due to isolation, were all predictors of mental health problems.’’

6. Page 2, Line 46:  Following sentence ‘ ‘Considering that people with autism spectrum  disorder….’’ must corrected as ‘ ‘Considering that people with ASD…..’’

7. Page 2, Line 62: The citation/citations needed for following sentence ‘‘The principal problem experienced by individuals with ASD was isolation resulting from the closure of schools and treatment facilities.’’

8. Page 2, Line 68: The citation/citations needed for following sentence ‘‘The advantages included the ease of using digital tools and support from parents and teachers.’’

9.  Page 2, Line 69-71: The citation/citations needed for following sentence ‘ ‘Among the disadvantages of online learning were difficulties in focusing, organizing self-study and self-discipline, loss of engagement, negative emotions (stress, fatigue, fear of passing exams), ang lack of direct contact with peers.’’

10. Page 2, Line 70: ang must be and.

11. Page 2, Line 74-76: What author want to mean is following sentence is unclear. Authors must rewrite this sentence ‘ ‘Previous studies of the potential effects of pandemic on the psychological well-being of adolescents with ASD are questionable.’’

12. Page 2, Line 80: It is hoped that must change with It is expected

13. Introduction, General: The main weakness of Introduction section is that author did not give any information about previous studies of students with ASD during he pandemic and their weakness that necessitate to this study. Thus, author must firstly add previous international and national studies and their weakness and then importance of their study. Authors did not give any information about these two points.

Method

14. Page 3, Line 100:  (N = 9)  in this N must be small and italic.

15. Page 3, Line 100:  female [K1] must be female [F1]

16. Participants Section, General:  Author must add mean age and its standard deviation in this section. Moreover, author must add sampling method to this section.

17. Page 3, Line 103:  I think complete families must be intact families.

18. Page 3, Line 118-120: Following information must remove to Participants section ‘ ‘Selection for the research group was via nonprobability sampling, and the criteria for selection were diagnosis of autism spectrum disorder, developed communication competence, and attendance at a state high school.’’

Results

19. Results section, General: Results section contain no problem.

Discussion

20. Page 9, Line 398: Please add beginning of sentence od Discussion following ‘‘Results of his study indicate that both……’’

21. Discussion, general: Practical implications of study findings are completely missing and must add discussion section with a paragraph.

Conclusion

22. Page 11, Line 496-497: Author indicated following limitation ‘ ‘The study did not reflect the daily life adolescents with level 2 and 3 autism spectrum disorder (according to the DSM-5).’’ However, did not give information about inclusion criteria about this. Thus, authors need to include level of autism as per DSM-5 to inclusion criteria.

Author Response

Dear Reviewer,

Thanks a lot for the time spent reviewing the article and for your useful comments and suggestions.   Please see the attachment.

Round 2

Reviewer 3 Report

Thanks for opportunity to rereview manuscript entitled ‘‘Psychosocial Functioning and the Educational Experiences of Students with ASD during the COVID-19 Pandemic in Poland’’ for International Journal of  Environmental Research and Public Health. They make a good job for improving quality of their manuscript. Authors revised the manuscript as I requested with a good will. In this form, Introduction reflects very well the previous studies and study aim, Method section and Result section is correct, and Discussion section adequately synthesis to previous study findings and current study results. Overall, I have no further comment regarding to manuscript. I congratulate to authors and wish them success on their future endeavors.